# Shared or Separate Representations? The Spanish Palatal Nasal in Early Spanish/English Bilinguals

**Sara Stefanich [1],* and Jennifer Cabrelli [2]** 

[1] Department of Spanish and Portuguese, Northwestern University, Evanston, IL 60208, USA
[2] Department of Hispanic and Italian Studies, University of Illinois at Chicago, Chicago, IL 60607, USA; cabrelli@uic.edu
* Correspondence: sara.stefanich@northwestern.edu

**Abstract:** The purpose of this study is to examine phonetic interactions in early Spanish/English bilinguals to see if they have established a representation for the Spanish palatal nasal /ɲ/ (e.g., /kaɲon/ *cañón* 'canyon') that is separate from the similar, yet acoustically distinct English /n+j/ sequence (e.g., /kænjn/ 'canyon'). Twenty heritage speakers of Spanish completed a delayed repetition task in each language, in which a set of disyllabic nonce words were produced in a carrier phrase. English critical stimuli contained an intervocalic /n+j/ sequence (e.g., /dɛnjɑ/ 'denya') and Spanish critical stimuli contained intervocalic /ɲ/ (e.g., /deɲja/ 'deña'). We measured the duration and formant contours of the following vocalic portion as acoustic indices of the /ɲ/~/n+j/ distinction. The duration data and formant contour data alike show that early bilinguals distinguish between the Spanish /ɲ/ and English /n+j/ in production, indicative of the maintenance of separate representations for these similar sounds and thus a lack of interaction between systems for bilinguals in this scenario. We discuss these discrete representations in comparison to previous evidence of shared and separate representations in this population, examining a set of variables that are potentially responsible for the attested distinction.

**Keywords:** heritage bilingualism; early bilingualism; Spanish; English; phonology; phonetics; speech production

## 1. Introduction

An overarching question in the field of bilingual phonology addresses the levels at and degree to which a bilingual's phonetic and phonological systems interact and how these interactions can be modelled within a theory of bilingual grammar. Bilingualism comes in many different forms, one of which is heritage speaker bilingualism. Herein, the term "heritage speaker" (HS) "refer[s] to any bilingual whose [first language] L1 (HL) was learned primarily at home as a minority language and whose [second language] L2 was learned primarily outside the home as the societal (majority) language" (Chang 2020, p. 2). As of result of this acquisition trajectory, heritage speakers' HL and majority language (ML) typically differ with regard to age and context of acquisition, frequency and context of usage, formal education, proficiency, and dominance (which often shifts from the HL to the ML once speakers reach school age), among other factors. These between-language differences yield a unique testing ground for the examination of how these factors modulate the nature of phonetic and phonological interactions in the bilingual mind.

Empirical investigations into the nature and degree of these interactions in heritage speaker phonologies have experienced an uptick over the last decade (see (Chang 2020) for a comprehensive review) and a survey of the growing body of research indicates that production patterns in the heritage language often lie between those attested in late L2 learners of the heritage language that are L1

speakers of the majority language (henceforth, L2ers) and in L1 speakers of the heritage language that have acquired the L2 as adults (henceforth, L1ers). Preliminary evidence suggests that segmental phenomena might be less vulnerable than suprasegmental phenomena to ML influence, albeit with substantial individual variability given the heterogeneity of HSs' language experience. Production data in the ML, on the other hand, although very limited, shows a clearer pattern of production that is typically indistinguishable from that of L1ers, particularly at the segmental level (e.g., Barlow 2014; Mayr and Siddika 2018; McCarthy et al. 2013). In the HL, much of the production research to date has examined segmental phenomena, with attention given to the representation of analogous sounds that are found in monolingual varieties of the ML and HL. That is, researchers have sought to determine whether HSs' production aligns with the baseline production data from L1ers or whether it shows influence from the ML. Existing research varies in outcomes between HL data that skew towards an L1 baseline (e.g., Chang et al. 2009, 2011; Lein et al. 2016) and those that do not (e.g., McCarthy et al. 2013; Ronquest 2012). This variability has been attributed to factors such as speaker generation and sociocultural factors (e.g., Nagy and Kochetov 2013), dominance (e.g., Amengual 2016, 2018; Shea 2019; Simonet 2014), proficiency (e.g., Shea 2019), age of ML acquisition (e.g., Barlow 2014; Cheng 2019), relative similarity between HL and ML sound(s) (e.g., Godson 2003, 2004; Yao and Chang 2016), and whether testing took place in monolingual versus bilingual testing mode (e.g., Amengual 2018; Simonet 2014; Simonet and Amengual 2020), among other things.

Most empirical studies report data solely from heritage speakers' HL, which prevents a direct comparison between HL and ML production that would allow for the verification of distinct HL and ML representations. While comparisons between heritage and baseline data are valuable in their own right, comparisons between the HL and ML are an important contribution to our understanding of heritage speaker phonology in that they allow us to determine the nature of the interaction of the HS's two phonologies. Specifically, we can determine for a crosslinguistic pair of sounds whether a speaker's system includes separate representations utilized in ML versus HL production or a single representation that relies on the production of both the ML and HL. The few studies that report direct comparisons suggest that heritage speakers maintain distinct representations in a shared phonetic space, even when the sound pair under investigation is considered to be similar—but acoustically distinct—in baseline varieties of the HL and ML (e.g., Amengual 2018; Chang et al. 2009, 2011; Knightly et al. 2003).

The studies that have compared HL and ML productions have tested one-to-one analogous sound correspondences between the HL and ML. In the current study, however, we examine a distinct crosslinguistic scenario, specifically the production of nasal sounds in heritage speakers of Spanish in the Midwest US. While the inventory of monolingual Spanish contains the palatal nasal phoneme /ɲ/ (e.g., *cañón* 'canyon' /kaˈɲon/), the inventory of monolingual English does not. However, an approximation exists in the form of the heterosyllabic phoneme sequence /n+j/ (e.g., 'canyon' /ˈkæn.jn/), which can be distinguished acoustically from the complex segment /ɲ/ via the duration and formant trajectories (e.g., Bongiovanni 2019).[1] Herein, we ask whether bilingual speakers of Spanish as the HL and English as the ML rely on distinct representations when producing these sounds in Spanish mode versus English mode.

Data from L1 English/(late) advanced L2 Spanish learners (Stefanich and Cabrelli 2016) have shown that advanced (late) L2 Spanish learners' productions patterned together in English and Spanish modes, and that this apparent shared category did not align with baseline (L1) Spanish data nor with the baseline English data provided by beginner L2 Spanish learners. That is, learners did not appear to create a novel L2 category when producing nonce words that were presented to them auditorily as /ɲ/; Stefanich and Cabrelli (2016) considered this shared intermediate representation to be a potential reflection of L2 influence on an early established L1 representation. This finding aligns with the

---

[1] A note on notation: although category representations are often represented in the literature using brackets, we use slashes when referring to phonemic inventories and representations in the speaker's grammar.

hypothesis that "similar" sounds in the L2 with an analogue sound in the L1 will be less salient in the input and, in turn, the learner will be less likely to create a novel category for it in the L2 (Flege 1995; Flege and Bohn 2020, but cf., e.g., van Leussen and Escudero 2015, whose (revised) Second Language Linguistic Perception (L2LP) model predicts that similar sounds will be less difficult than different sounds). In light of these L2 data, we examine herein whether early bilinguals' data align with those of their advanced L2 counterparts, or whether these speakers' qualitative and quantitative differences in language experience yield separate representations when in Spanish versus English mode.

After an overview of the relevant nasal consonant inventory in Spanish and English and their acoustic properties, we present the research question and predictions specific to this crosslinguistic scenario. Then, we detail the methods and the results, followed by a discussion. The results from a delayed repetition task administered in separate Spanish and English modes suggest that early bilinguals rely on distinct representations in each mode; the acoustic data indicate that they produce a complex segment in Spanish mode versus a two-segment sequence in English mode. This outcome thus suggests a lack of interaction between systems for these bilinguals in this case, despite the crosslinguistic proximity between English /n+j/ and Spanish /ɲ/. We discuss these discrete representations in comparison to previous evidence of merged versus separate representations in this population and examine the variables that are potentially responsible for the distinction.

### 1.1. Nasal Consonants in Spanish and English

Spanish has three nasal phonemes that contrast by place of articulation in syllable onset position: bilabial /m/, alveolar /n/, and alveolopalatal /ɲ/ (Díaz-Campos 2004; Recasens 2013) (1).

1.  /m/    *cama*    /ˈka**m**a/    'bed';
    /n/    *cana*    /ˈka**n**a/    'gray hair';
    /ɲ/    *caña*    /ˈka**ɲ**a/    'cane'.

The palatal nasal /ɲ/ is the least frequent phoneme in Spanish (Melgar de González 1976) and is a complex segment comprised of an alveolar nasal element followed in succession by a palatal glide element (Martínez Celdrán and Planas 2007; Massone 1988) posited to be phonologically associated with the nasal segment (e.g., Colina 2009).

Although English lacks a phonemic palatal nasal (the inventory is limited to /m n ŋ/), a similar but heterosyllabic /n+j/ sequence is found in words such as *canyon*, *onion*, and *lanyard* (2).

2.  *canyon*    /ˈkænjn/;
    *onion*    /ˈʌnjn/;
    *lanyard*    /ˈlænjɹd/.

While there are no published data on the acoustic quality of the English /n+j/ sequence to inform the acoustic analysis parameters that distinguish the complex segment /ɲ/ from the discrete segments of /n+j/, a similar (albeit tautosyllabic) sequence is found in Spanish in the surface form of words such as *uranio* /uɾanjo/ 'uranium' and has been investigated acoustically. In Spanish, both /n+j/[2] and /ɲ/ are composed of a combination of a nasal element and a palatal glide element; /ɲ/ is a single complex segment in which the glide element is said to be "partial" (versus a "full" element in /n+j/), Martínez Celdrán and Planas 2007). On the other hand, /n+j/ is a sound sequence in which a "full" glide element is an independent segment. Phonologically, the sound sequence is hypothesized to differ from /ɲ/ in that the glide element in /n+j/ is associated with the following vowel, forming a complex nucleus (e.g., Colina 2009); we assume this to be the case for English /n+j/ as well.

Despite their commonalities, the pair has been found to be distinguished acoustically in word pairs such as *uranio* /uɾanjo/ 'uranium' and *huraño* /uɾaɲo/ 'unsociable' (see Bongiovanni 2019 for a

---

[2]   We employ this phonemic notation following Bongiovanni (2019), recognizing that the glide in this sound sequence in Spanish is not phonemic and that this notation conflates phonetic and phonological representations.

review of acoustic and articulatory evidence). In Bongiovanni's (2019) study of /n+j/ and /ɲ/ production in Buenos Aires Spanish, an analysis of the vocalic portion[3] following the nasal consonant supported the phonological association of the glide to the nasal consonant in /ɲ/ (i.e., ɲV) and the glide to the vowel nucleus in /n+j/ (i.e., njV). Specifically, the gestural difference in the timing and degree of lingual-palatal contact reported in studies such as Recasens (2013) was acoustically evident in formant contour trajectories (i.e., the rise of F2 and the decrease in F1 in /n+j/), the timing at which F1 minimum and F2 maximum were reached (i.e., the timing for /ɲ/ should be earlier), and the duration of the vocalic portion (predicted to be longer in /n+j/ given its status as part of a complex nucleus). Although there is no crosslinguistic research that examines /n+j/ in English versus Spanish, given the heterosyllabic nature of English /n+j/ versus the tautosyllabic /n+j/ in Spanish, it is logical to predict that the glide will be even more clearly associated with the following vowel in English. These predicted differences are visible in the spectrograms and waveforms in Figures 1–3, which are taken from a participant's productions of the nonce item 'denya' in English mode (Figure 1) and 'deña' in Spanish mode (Figure 2), with 'dena' in Spanish mode (Figure 3) as a point of comparison. In the current study, we follow Bongiovanni (2019) and measure duration and formant contours as a correlate of the phonological association of the glide element. As she notes, reporting data from both measures will allow for the confirmation of the reliability of each measure and the avoidance of the overgeneralization of data based on a single measure.

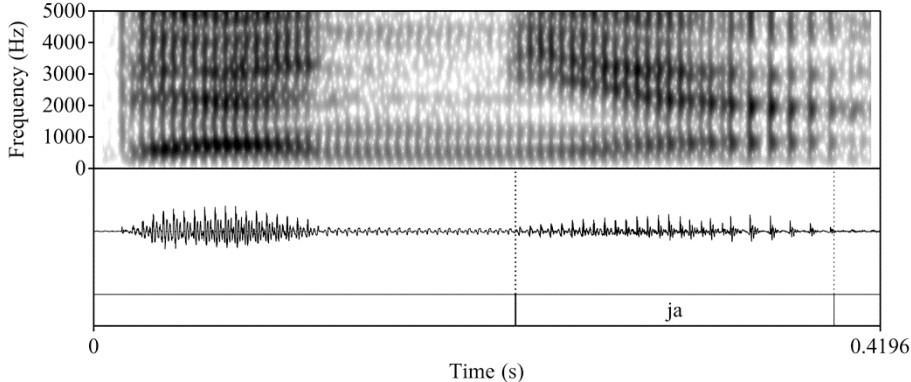

**Figure 1.** Waveform and spectrogram of a participant's production of /dɛnjɑ/ 'denya' in English mode.

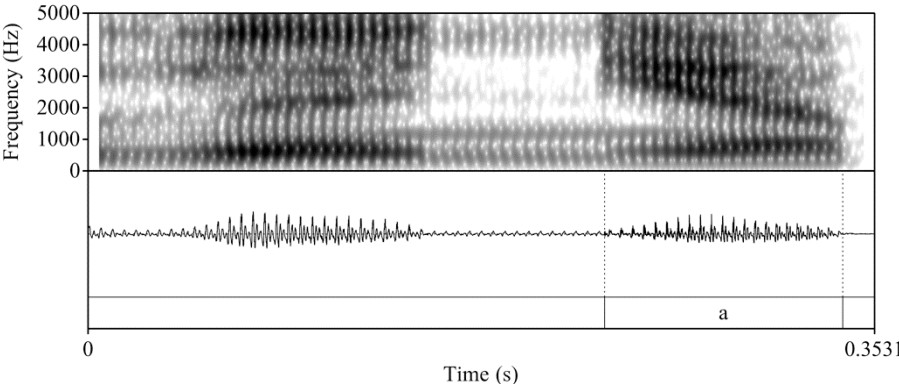

**Figure 2.** Waveform and spectrogram of a participant's production of /deɲa/ 'deña' in Spanish mode.

---

3     In light of the unreliability of acoustic analysis of nasal consonants (see, e.g., Fujimura 1962, cited in Bongiovanni 2019, p. 4), Bongiovanni (2019) limited her analysis to the following vocalic portion.

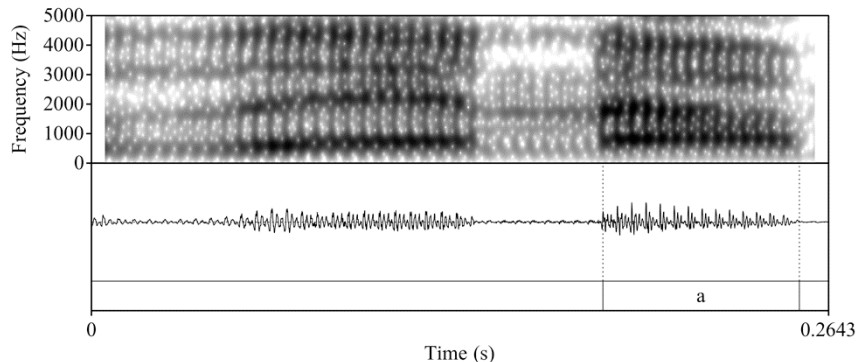

**Figure 3.** Waveform and spectrogram of a participant's production of /dena/ 'dena' in Spanish mode.

*1.2. Research Question and Predictions*

The research question that drives this study is the following: do heritage speakers of Spanish evidence distinct representations in their productions of /ɲ/ when in Spanish mode and /n+j/ when in English mode? This is an exploratory question with three possible outcomes: The first is that the differences between language mode in duration and/or formant contours will reveal that these speakers maintain distinct representations. In the case that the quality of these differences patterns with the acoustic parameters associated with /ɲ/ versus /n+j/, such an outcome would be suggestive of implicit knowledge of the distinct single complex segment /ɲ/ in Spanish versus the segment sequence of /n+j/ in English. However, it is wholly possible that speakers will rely on cues other than those reported in the baseline literature, as seen in work on within-language contrasts (e.g., Amengual 2016). The second is that there are no between-language acoustic differences and that the duration and formant contour data skew towards the acoustic description of /n+j/ (i.e., a two-segment sequence rather than a complex segment). The third, like the second, is a lack of between-language differences, but with data that pattern with the acoustic description of /ɲ/. In the latter two cases, it will be necessary to consider what might drive the privileged status of one representation over the other. In terms of predictions, while this is the first study to our knowledge to examine a complex segment compared with a two-segment sequence, we can look to the minimal research that directly compares HL and ML segmental data. As noted in Section 1, when limited to HL data, it is difficult to draw strong conclusions about the interaction of the HL and ML without ML data as a point of comparison. We can of course predict that, if the HL is baseline-like, and knowing that HSs typically are baseline-like in the ML (see Chang 2020, p. 10 for discussion), then they have two representations. Our question, however, is not how close the ML or HL production is to a baseline, but rather whether the speakers' production patterns are different in English mode versus Spanish mode. As we have mentioned, the few studies that directly compare HL and ML data indicate distinct representations in a shared phonetic space, which yields the prediction that the HS in the present study will distinguish acoustically between /ɲ/ and /n+j/ in production. In the case that they do not, we predict that the production will skew towards the reported English pattern, given that (a) the HSs are largely English dominant and (b), overall, segmental differences in the ML when compared to the ML "norm" are small and variable, without clear evidence to date that any measured differences are perceivable (Chang 2020, p. 10).

## 2. Materials and Methods

*2.1. Participants*

Twenty Spanish/English bilinguals participated in this study. At the time of the study, all the participants were undergraduate students living in the Chicagoland area. The participants ranged in age from 18–25 ($M = 21.05$, $SD = 1.47$). All the participants reported learning Spanish before the age of 3 ($M = 0.25$, $SD = 0.79$) and English before the age of 8 ($M = 3.30$, $SD = 2.73$). Specifically, six participants reported learning Spanish and English since birth, whereas thirteen reported learning Spanish before

English and one participant learned English before Spanish. We estimate that the majority of these participants are second-generation HS, as approximately 85% of the Spanish HS at the institution where the data were gathered are second-generation speakers (Potowski 2020). The participants reported that, for any given week, they use more English than Spanish with friends and at school/work but more Spanish than English with family (Table 1).

**Table 1.** Mean percent of language use by domain.

| | English | | Spanish | |
|---|---|---|---|---|
| | *M* | *SD* | *M* | *SD* |
| Friends | 0.82 | 0.15 | 0.18 | 0.15 |
| Family | 0.41 | 0.25 | 0.59 | 0.23 |
| School/Work | 0.78 | 0.13 | 0.20 | 0.14 |

As a proxy for language dominance, the participants completed the Bilingual Language Profile (BLP, Birdsong et al. 2012), a bio-linguistic questionnaire which uses the participants' responses to provide a language dominance score on a scale of −218 (Spanish dominant) to 218 (English dominant), with "0" indicating a "balance" between the two languages.[4] The majority of our participants scored on the English side of the scale (*n* = 17), with a range of scores from −22.7 to 88.6 (*M* = 43.56, *SD* = 35.35); the three participants who scored on the Spanish side of the scale fell very close to the balanced zero point. As part of the BLP, the participants rated their Spanish and English proficiency in speaking, understanding, reading, and writing on a scale from 1 (not very well) to 6 (very well) (Table 2). In addition to self-rated proficiency, the participants completed a 50-item written Spanish proficiency assessment composed of portions of the Diploma of Spanish as a Foreign Language (DELE) and Modern Language Association (MLA) assessments commonly administered in heritage research (e.g., Keating et al. 2016; Leal et al. 2015). Our participants averaged a score of 35.50 (*SD* = 7.80) on the written assessment. Dominance and written proficiency were found to be weakly negatively correlated (r(18) = −0.32, *p* = 0.175).

**Table 2.** Self-reported proficiency (scale 1–6).

| | English | | Spanish | |
|---|---|---|---|---|
| | *M* | *SD* | *M* | *SD* |
| Speaking | 5.75 | 0.55 | 4.35 | 1.27 |
| Understanding | 5.95 | 0.22 | 5.00 | 0.97 |
| Reading | 5.85 | 0.87 | 4.05 | 1.32 |
| Writing | 5.75 | 0.44 | 3.65 | 1.27 |

Heritage speaker populations have been shown to be heterogeneous in terms of language experience and use (e.g., Montrul and Polinsky 2019), and the sample in the current study is no exception. We acknowledge the heterogeneity of these Spanish/English bilinguals in terms of age of acquisition, proficiency, language use, and language dominance and address a number of these factors as they relate to the outcomes in our discussion (Section 4).

*2.2. Materials and Procedure*

The experiment consisted of Delayed Repetition Tasks (e.g., Trofimovich and Baker 2006) in English mode and Spanish mode. Each task included 40 trials (10 critical, 10 control, 20 distractor). Each trial

---

[4]  Following authors such as Birdsong (2016) and Solis-Barroso and Stefanich (2019), we recognize the gradient nature of the different dimensions of dominance and treat the variable as scalar rather than categorical.

was composed of a target nonce word presented auditorily within the carrier phrase 'I'm saying ___ to you' in English and its equivalent *Digo X para ti* in Spanish. A 1000 ms silent pause was then followed by the spoken prompt "What are you saying to me?" in English or the equivalent *¿Qué me dices?* in Spanish, which prompted the participant to produce the original phrase. Items in both tasks had penultimate stress and were phonotactically licit in the respective language presented. Critical items followed a (C)CV$^1$n.jV$^2$ (English) or (C)CV$^1$.ɲV$^2$ (Spanish) structure and were counterbalanced in each language with 10 control items containing the alveolar nasal /n/ in a (C)CV$^1$.nV$^2$ structure.[5] Across critical and control conditions, V$^1$ was a mid or low vowel (/ɛ/ or /ɑ/ in English and /e/ or /o/ in Spanish; V$^2$ was /a/ in Spanish and /ɑ/ in English. The 20 distractors followed the same general (C)CV.CV structure as the control and critical items. The item composition in the two tasks is summarized in Table 3; the full set of stimuli is in Appendix A. English stimuli were recorded by a phonetically trained female native speaker of Midwest American English; Spanish stimuli were recorded by a phonetically trained female native speaker of Northern Peninsular Spanish.

**Table 3.** Composition of nonce stimuli in the English and Spanish delayed repetition tasks.

|  | *n* | English | Example | | Spanish | Example | |
|---|---|---|---|---|---|---|---|
| Critical | 10 | (C)CVn.ja | /dɛnjɑ/ [ˈdɛn.jə] | 'denya' | (C)CV.ɲa | /deɲa/ [ˈde.ɲa] | *deña* |
| Control | 10 | (C)CV.na | /dɛnɑ/ [ˈdɛ.nə] | 'denna' | (C)CV.na | /dena/ [ˈde.na] | *dena* |
| Distractor | 20 | (C)CV.CV | /lɛkɑ/ [ˈlɛ.kə] | 'lecka' | (C)CV.CV | /meba/ [ˈme.βa] | *meba* |

Trials were presented using E-prime 2.0 (Psychology Software Tools, Inc., Pittsburgh, PA, USA); audio stimuli were presented over Sennheiser HD-280 PRO (Sennheiser, Wedemark, Germany) headphones through a MOTU Ultralite mk3 interface (MOTU, Cambridge, MA, USA). Recordings took place in a sound-attenuated booth using a head-mounted Shure SM 10A (Shure Inc., Niles, IL, USA) dynamic microphone and a Marantz PMD 661 solid-state recorder (Marantz Corp., Kawasaki, Japan) at a 44.1 kHz sampling rate.

Data were collected in a single session that consisted of separate English and Spanish session modes; the mode order was counterbalanced across participants. All the participants provided informed consent following University of Illinois at Chicago IRB protocol 2015-0040 prior to data collection. The English mode session began with a 10 min interview with the participant to establish the language mode. The participants then completed the English delayed repetition task and the BLP. The Spanish mode session consisted of a 10 min interview, the Spanish delayed repetition task, and the written proficiency assessment.

*2.3. Analysis*

2.3.1. Acoustic Analysis

Following the literature presented in Section 1.1, this study examines the duration and formant contours of the vocalic portion following the nasal segment. To that end, sound files were segmented and analyzed in Praat [6.1.16] (Boersma and Weenink 2019). The theoretical ceiling of tokens was 600, or 30 per speaker (10 Spanish critical, 10 English critical, 10 Spanish control). One participant's data was excluded from analysis due to a lack of discernible impressionistic difference between /n/ and /ɲ/ in Spanish. Further, an additional 14 tokens were removed due to non-target productions (participants

---

[5]　Spanish alveolar data are reported for contextual comparison; we have excluded the English alveolar data, as they are not relevant to the research question.

skipping, repeating, or producing different segments), creaky voice, or background noise, for a final total of 556 tokens.

During segmentation, the onset of the vocalic portion was determined by the visual presence of an abrupt change in formant structure and frequencies, and the offset was determined by a breaking up of the formant structure and a loss of energy and periodicity in the waveform (Ladefoged 2005). Following Bongiovanni (2019), boundaries between formant transitions or between the glide and the vowel /a/ were not marked.

Once segmentation was completed, measurements were extracted via scripts (Hirst 2012 for automatic duration measurements; McCloy and McGrath 2012 for semi-automatic formant measurements). Formant measurements were taken at 20 points within the vocalic portion (every 5%); 5.2% of the data were manually corrected where it was evident that there were formant tracking errors with the Praat script.

### 2.3.2. Statistical Analysis

For the duration of the vocalic portion, a linear mixed model (LMM) was fit to the data (measured in ms)[6] using the MIXED procedure in SPSS 26 (IBM Corp. 2019) with a fixed effect of language mode (English, Spanish). The random effects structure (RES) was the maximal structure supported by the data (Barr et al. 2013) and included by-subject and by-item intercepts.

For the formant structure of the vocalic portion, we followed the analysis laid out by Bongiovanni (2019). The formant values were transformed to Bark units, and Smoothing Spline ANOVA (SSANOVA) were fit to the data (time points and corresponding Bark units at each time point) in R, version 4.0.2 (R Core Team 2020), with the gss package. Here, a smoothing spline fits a smooth curve to the observations and the SSANOVA determines whether the curves in question are statistically different from one another (i.e., whether their confidence intervals overlap). As in previous research (e.g., Bongiovanni 2019; Kirkham 2017; Nance 2014; Simonet et al. 2008), we limit our report to the graphical representations of the SSANOVA.

## 3. Results

### 3.1. Duration Results

A visual representation of the duration data is presented via the boxplot in Figure 4; as predicted, the vocalic portion of /n+j/ produced in English mode was longer than that of /ɲ/ produced in Spanish mode. The LMM yielded a significant main effect of language ($F(1,41.942) = 70.524$, $p < 0.001$); a Bonferroni post-hoc comparison showed that the vocalic portion for the English /n+j/ ($M = 169.50$ ms, SE = 5.13, CI [159.17,179.84]) was longer than for the Spanish /ɲ/ (107.39 ms, SE = 5.33, CI [96.63,118.16], $p < 0.001$). Hedges' g was calculated as a measure of the effect size on the raw means and standard deviations (English $M = 137.75$ ms, $SD = 42.95$ ms; Spanish $M = 85.78$, $SD = 17.92$) and yielded a large effect size of 1.51 (according to Plonsky and Oswald 2014, for within-subject comparisons). This outcome aligns with the predicted crosslinguistic difference and is indicative of distinct representations in Spanish and English.

---

6    To determine the effect of individual differences in speech rate on the outcome, a separate model was fit to z-score-transformed data; the model yielded the same main effect of language ($F(1,41.942) = 70.524$, $p < 0.001$). For ease of interpretation, we report the duration data herein in ms.

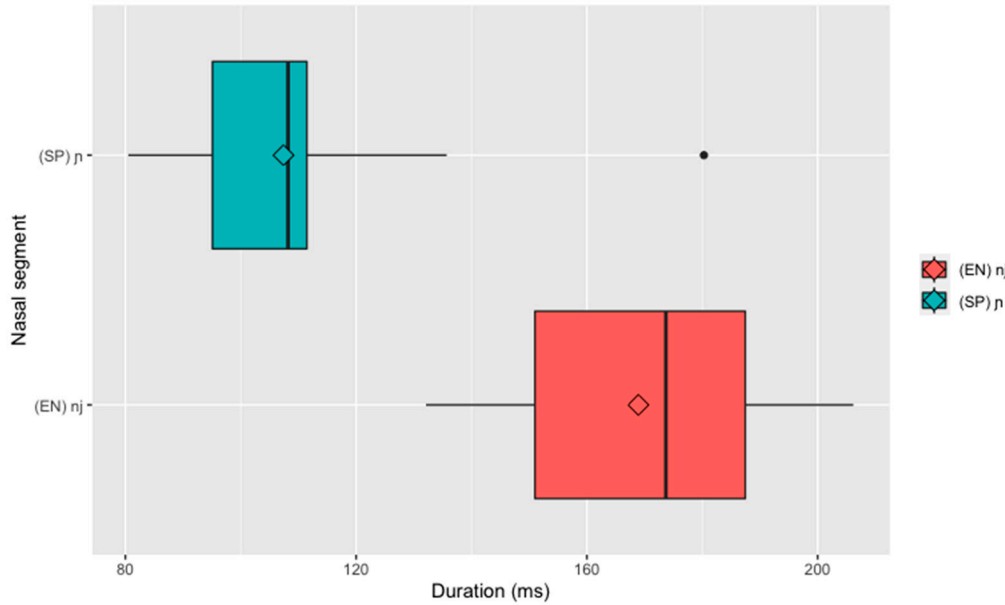

**Figure 4.** Duration of the following vocalic portion of /n+j/ produced in English mode and /ɲ/ produced in Spanish mode Note: "Nasal segment" refers to /ɲ/ and /n+j/; diamonds represent duration means.

### 3.2. Formant Structure Results

Recall that, with SSANOVA, statistical significance is indicated by non-overlapping confidence intervals plotted around the data-generated formant curves. The acoustic differences between /ɲ/ and /n+j/ are predicted to take the form of a lower F1 and a higher F2 for /n+j/ than /ɲ/. Keeping these predictions in mind, the results of the SSANOVA are presented in Figure 5.

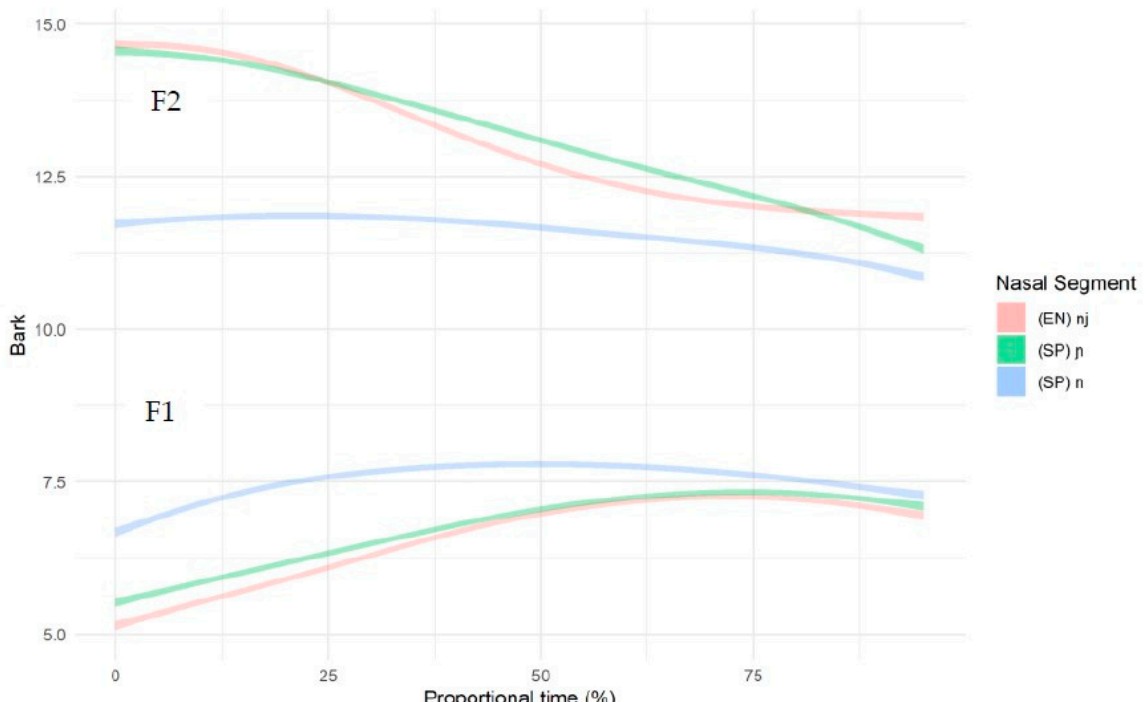

**Figure 5.** Smoothing Spline ANOVA of formant trajectories by nasal segment.

For F1, the confidence intervals of the /ɲ/ and /n+j/ curves do not overlap between the 0% and 40% points, after which they run adjacent to one another between the 40% and 100% points with a

slight overlap between 50% and 75%. For F2, although there is no overlap between 0–20% and 30–80%, the intervals for /ɲ/ and /n+j/ overlap at two points (at roughly 25% and 85%), illustrating a steeper negative slope for /ɲ/ versus /n+j/. These formant readings follow the predicted shapes, with a lower F1 and higher F2 for /n+j/ than for /ɲ/, although these differences are small, with a maximum of between 0.55 and 0.61 Bark at their most different. This difference falls below the assumed just-noticeable difference (JND) threshold of 1 Bark unit, which we address in the discussion in terms of whether this difference is perceivable. In contrast, there is zero overlap in the confidence intervals for Spanish /n/ versus /n+j/ and /ɲ/, with differences that exceed the JND threshold. For Spanish /n/ versus English /n+j/, differences in F1 range from 1.15 to 3.17 Bark and in F2 from 1.61 to 3.68 Bark at their most different. For Spanish /n/ versus /ɲ/, the differences in F1 range from 1.00 to 3.01 Bark and in F2 from 1.48 to 3.16 Bark at their most different.

## 4. Discussion

### 4.1. Summary

This study investigated the speech production of a group of Spanish heritage speakers with English as the ML to determine whether their production patterns are acoustically distinct when producing /ɲ/ in Spanish mode versus /n+j/ in English mode. The between-mode differences, which we took to indicate separate representations in a shared phonetic space, were determined via two acoustic indices: (1) the duration of the vocalic portion following the nasal segment (hereafter, FV) and (2) the formant trajectories of the same FV. Acoustic differences were predicted to present in the form of (a) a longer FV duration for /n+j/ than /ɲ/ and (b) formant trajectories in which the /n+j/ evidenced a lower F1 valley and higher F2 peak than /ɲ/, as indicated by non-overlapping formant contours.

The results from the duration analysis confirmed the prediction—the FV for /n+j/ was significantly longer than the FV for /ɲ/. The results from the SSANOVA also fell in line with the expected predictions for the differences between the formant contours; the formant trajectory of the FV for /n+j/ diverged from that of /ɲ/ for portions of the vowel and evidenced a lower F1 and higher F2. Taken together, these results suggest that this group of Spanish HS draws on distinct representations when producing /ɲ/ in Spanish mode versus /n+j/ in English mode. That is, despite the similarities between /n+j/ and /ɲ/, for these participants there is no evidence of interaction between the two phonological systems in this particular case.

### 4.2. Separate Representations and Age of Acquisition

A lack of interactions between phonological systems suggests that these early bilinguals had sufficient input to develop the representation for the sound they produce when in Spanish mode, despite the fact that /ɲ/ is the least frequent phoneme in the Spanish inventory (Melgar de González 1976). Moreover, they appear to have maintained the representation even after (in most cases) switching dominance to the ML and developing a representation of /n+j/ that is evident in the English mode data. These data align with previous findings that suggest that phonological systems are less susceptible to interaction at the segmental level vs. the suprasegmental level (see Chang 2020 for review). Our findings also add to those that have reported distinct representations of sounds in a shared phonetic space that are similar but acoustically different (e.g., Amengual 2018; Chang et al. 2009, 2011; Knightly et al. 2003, but cf. e.g., Godson 2003; Kang et al. 2016).[7]

---

[7]   One factor that may contribute to why the data do not evidence merged categories, such as those in the voiced stop data in Kang et al. (2016) and the acoustically similar vowel data in Godson (2003), is that some similar crosslinguistic pairs might be "easier" to keep separate. Recall from Section 1.1 that Spanish also has a /n+j/ sequence that contrasts with /ɲ/ in pairs, such as *uranio* /uɾanjo/ 'uranium' and *huraño* /uɾaɲo/ 'unsociable'. Although the only experimental data on this contrast we are aware of is from Buenos Aires Spanish, in which there is a near-merger of /ɲ/ and /nj/, Bongiovanni (2019) found that, even in that case, while the participants did not accurately perceive the difference, their productions were acoustically distinct despite the contrast's low functional load. We posit that one possibility is that the early Spanish bilinguals in this

Interestingly, however, the attested distinction was not found in the production of advanced L1 English/L2 Spanish learners (Stefanich and Cabrelli 2016) in a study that employed the same task. Instead, Stefanich and Cabrelli determined that the advanced L2ers' productions were representative of a merged/hybrid category that was used when producing /ɲ/ in Spanish mode and /n+j/ in English mode. Because these advanced L2ers' data in English mode were different from a group of beginner L2ers' data (used as a proxy for the L2ers' L1 baseline in light of their minimal exposure to L2 input), Stefanich and Cabrelli (2016) posited the development of a hybrid category. While it is important to note that the acoustic index used was the duration of the nasal segment and thus not directly comparable[8], the contrast leads us to the following question: which factors might possibly yield interaction between phonological systems for adult L2ers but not for HSs? A primary difference between these two groups of bilinguals lies in their age of acquisition (AoA).[9] While all the HS participants reported having acquired Spanish before the age of three and English before the age of eight, the mean L2 Spanish AoA was 14.5 (*SD* = 4.21). A substantial body of research suggests a critical period for phonology around 5 years old (e.g., Barlow 2014; Flege et al. 1999; Newport et al. 2001; Scovel 2000, but cf. work that posits a later critical or sensitive period, e.g., DeKeyser 2012, or a lack of one overall, e.g., Abrahamsson and Hyltenstam 2009), and Barlow (2014) indicates that L1 influence on the L2 is more likely after the cutoff age. For instance, in her analysis of Spanish/English laterals, Barlow (2014) found that late bilinguals (AoA > 6) showed evidence of the (English-like) allophonic distribution of [l]~[ɫ] in English and in Spanish compared with early bilinguals (AoA < 5), who only showed it in English. That is, the late bilinguals evidenced interaction between systems (influence from L2 → L1), whereas the early bilinguals did not. Given that this outcome patterns with our HS and L2 results, we suggest that age of acquisition is a good candidate to be a predictor of whether interactions will occur. To confirm this hypothesis, however, we will need to compare groups with early versus late AoA that are matched (as closely as possible) in Spanish proficiency and dominance.

### 4.3. Individual Variation

While the group results indicate that these heritage speakers have separate representations, there is substantial variability in how this distinction is acoustically realized. While it could be the case that these individual patterns are simply noise in the sample, it is valuable to consider whether certain factors previously reported to condition bilingual speech patterns might explain some of the attested variability. Specifically, we discuss dominance, proficiency, and individual differences related to perception.

Although the purpose of this study was not to *a priori* examine the effects of proficiency and dominance, an analysis of the individual participants' data as they relate to the measures of dominance and proficiency used allows us to examine any trends in the relationship between them and the FV duration and formant contours. Figures 6 and 7 illustrate the difference in duration (in ms) between /n+j/ and /ɲ/ for each participant by proficiency score and dominance score, respectively.

---

study successfully developed these separate representations early on, and that doing so facilitated the acquisition of the /n+j/ sequence in English. Comparisons of the /n+j/ productions in English versus Spanish mode will inform whether there is a single representation of the /n+j/ sequence or two, thus providing a more complete picture of the crosslinguistic relationship of these similar sounds.

[8] Reanalysis of the L2 data from Stefanich and Cabrelli (2016), which will include the measurement of the same acoustic indices (FV duration and formant contours), is in progress.

[9] Both groups of bilinguals are overall English dominant, strengthening our conclusion that it is not merely language dominance alone that contributes to the interaction (or lack thereof) between systems. Further, given that language dominance is thought to be fluid and changeable across a bilingual's lifespan (e.g., De Houwer 2011), it makes sense that dominance would not be a determining factor in system interaction.

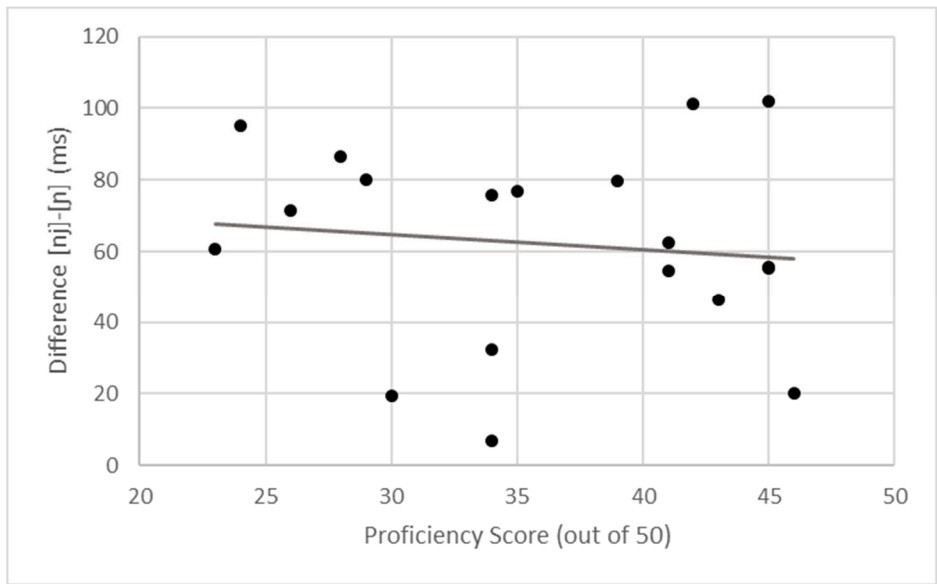

**Figure 6.** FV duration difference by Spanish proficiency score.

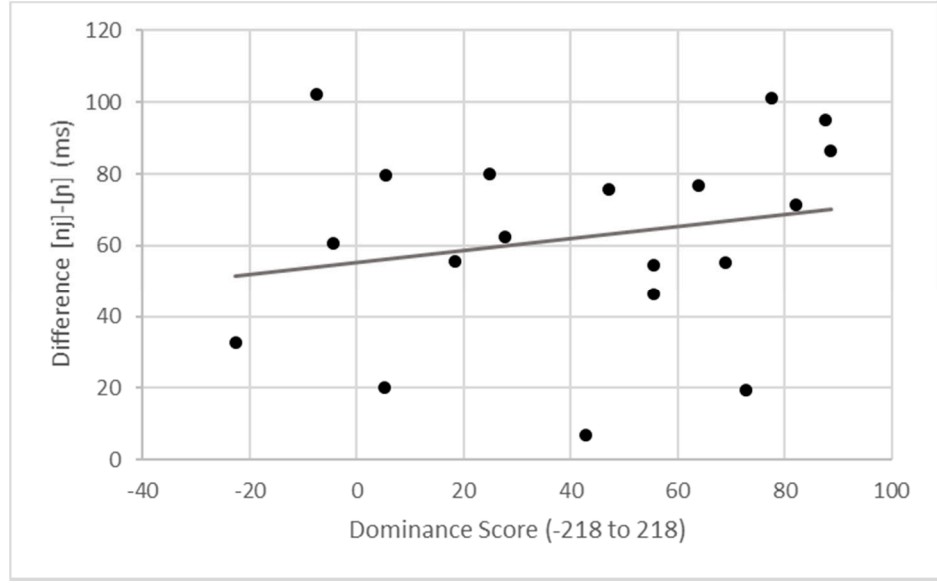

**Figure 7.** FV duration difference by language dominance score.

Previous research on heritage phonology has found that proficiency and dominance in the HL can at least partially account for individual variation, including the relationship between measures of these constructs and the robustness of a distinction (at least for within-language contrasts—see, e.g., Amengual 2016 for dominance, and Shea 2019, who examined dominance and proficiency and found proficiency to be a stronger predictor). In the current scenario in which we examine between-language distinctions, we might predict that bilinguals with a lower Spanish proficiency score would show smaller differences in FV duration and formant contours than those with higher scores. Regarding dominance, we might expect that bilinguals that fall closer to the balanced zero-point on the dominance scale to evidence greater distinctions than those bilinguals who fall towards the ends of the scale, whose representations would be predicted to skew towards the respective languages.

With respect to the Spanish language proficiency measure reported in Section 2.1, there does not seem to be a discernible pattern; the lack of a relationship visible in Figure 6 is supported by a weak negative correlation (r(18) = −0.11, *p* = 0.639). If proficiency as measured here played a larger

role, we might expect to find a strong positive correlation whereby, as proficiency in the HL increases, so does the difference in duration, maximizing the distance between productions in English versus Spanish modes. This lack of a relationship also seems to be the case for the language dominance measure; the BLP score and duration difference are only weakly correlated (r(18) = −0.21, *p* = 0.387). Moreover, if we examine dominance as binary, the three Spanish-dominant participants did not produce durational differences that were substantially larger or smaller than the English-dominant participants, nor did we find any clustering of duration differences around the "balanced" point on the scale.

Turning to the formant contours, the individual SSANOVA (Appendix B), unlike the duration differences, do show differences between participants that can be grouped into four patterns. Of the 19 participants, eight participants' individual splines mimic that of the group-level pattern, i.e., a lower F1 and a higher F2 for /n+j/ than /ɲ/ (Pattern 1). One participant diverges from the group pattern, with /ɲ/ having a lower F1 than /n+j/ (Pattern 2), and seven participants diverge from the group pattern, with /ɲ/ having a higher F2 than /n+j/ (Pattern 3). Lastly, three participants diverge from the group pattern for both F1 and F2. In this latter case, the pattern mirrors that of the group, exhibiting a lower F1 and a higher F2 for /ɲ/ than /n+j/ (Pattern 4). Thus, the majority either align with the group pattern or produce a more fronted[10] FV in /ɲ/ than /n+j/[11]. Figures 8 and 9 present the proficiency and dominance scores by grouped spline pattern to visualize any potential relationships between formant contour patterns and proficiency and dominance scores.

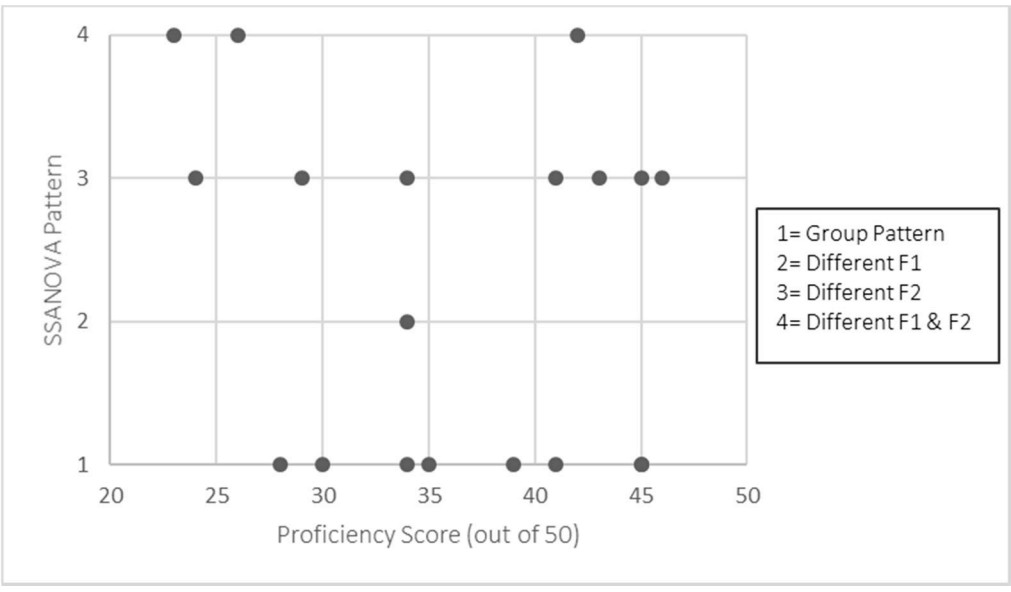

**Figure 8.** Smoothing Spline pattern type by Spanish proficiency score.

Upon examination of Figures 8 and 9, it does not seem to be the case that either Spanish language proficiency or language dominance as measured in this study can explain the individual patterns. That is, it does not appear that participants with lower Spanish proficiency scores differed from the group pattern more frequently than those with higher scores, or vice versa. Further, the three Spanish-dominant participants did not behave differently from the English-dominant participants. Thus, taking into account the individual analyses for both the FV duration and formant contours,

---

[10] As pointed out by an anonymous reviewer, another interpretation of the higher F2 values could be that these speakers are producing a more constricted dorsopalatal realization, given that dorsopalatal constriction narrowing and F2 are positively correlated.

[11] As evident in Appendix B, while the individual data fall into the four patterns, there is variation in the degree of spline overlap (i.e., acoustic distance). Without a principled way to quantitatively determine acoustic difference in the formant trajectories, however, we limit our discussion to the categorical patterns and include the individual SSANOVA for readers' reference.

there is no evidence that written Spanish language proficiency score nor language dominance score explain the observed individual variation. Why not? With respect to dominance score, there might not be sufficient variation between participants: recall that the BLP scale is from −218 (Spanish) to 218 (English); our participants' scores are concentrated between −22.7 and 88.6 ($M$ = 43.56, $SD$ = 35.35). Further, we only had three participants with scores on the Spanish side of the scale. Thus, it could be the case, given a wider range of dominance scores and a larger sample, that clearer patterns might emerge with respect to language dominance. Finally, given that dominance can be such an elusive construct, it is possible that a different proxy for dominance could reveal a relationship in the data that the BLP does not; Solis-Barroso and Stefanich (2019) evaluated a set of assessments that were completed by a single group of heritage Spanish bilinguals in Chicago and found different categorization patterns (dominant in Language A, dominant in Language B, "balanced") depending on the assessment, particularly when it came to bilinguals that fell close to the balanced point of a scale.

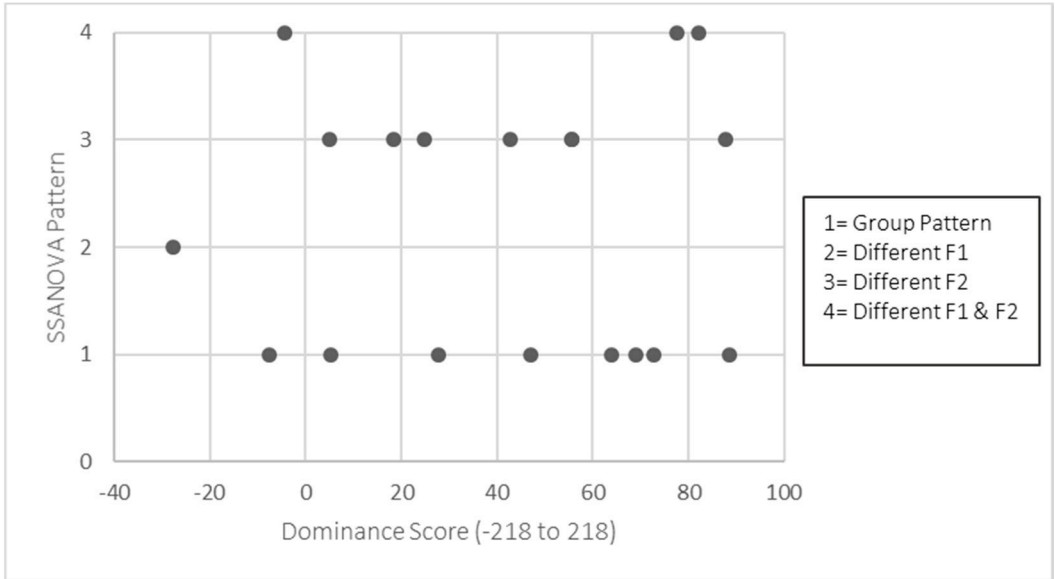

**Figure 9.** Smoothing Spline pattern type by dominance score.

For the Spanish proficiency score, unlike the dominance score, our participants show a wider range of scores from 23 to 46 ($M$ = 35.50, $SD$ = 7.80), and therefore one might expect to see a difference between those bilinguals with higher scores versus lower scores. However, in our case we see no such difference. Recall that proficiency was measured by a 50-item written measure, which we recognize is not ideal for several reasons, the most relevant of which are that (a) the focus of the study is speech production, and (b) a written assessment disadvantages heritage speakers without substantial formal education in the HL, particularly when the assessment is targeted at speakers of Peninsular Spanish rather than Mexican or US Spanish. Going forward, it would be ideal to employ a measure of oral proficiency (e.g., accent ratings, elicited imitation tests) or a variety of assessments that can be used to formulate a composite score or evaluated independently, such as the set of monologues, picture naming task, and vocabulary assessment used in Shea (2019). It could be the case that, with more direct measures of oral proficiency or a more global evaluation, patterns would more readily emerge in examining interactions between systems.[12]

---

[12]　We also note that we only measured our participants' proficiency in the heritage language (here, Spanish), as our participants attended school and are dominant in the majority language (here English). Future research could also measure proficiency in the majority language in addition to that of the heritage language to see what patterns might surface.

A final consideration when accounting for the attested variation is that of perception. While we have measured variables that have been reported to acoustically distinguish /n+j/ and /ɲ/, it remains to be verified whether these are the acoustic cues that these speakers attend to in the input, even in the cases in which the contour differs beyond the JND threshold. That is, it is wholly possible that these speakers (as a group, or individually) attend to different acoustic cues in the input and that these cues are what they use to distinguish in production, as well. Future research will examine the bilingual perception of the /ɲ/ and /n+j/ in Spanish and English modes, comparing stimuli which vary in the cue(s) (single cues and combination of cues) that are manipulated and held constant. Once we know that these bilinguals perceive the difference between /ɲ/ and /n+j/ and what cues they attend to in the input, an experiment can be designed to isolate production of those cues to confirm separate representations for /n+j/ versus /ɲ/.

### 4.4. Conclusions

In this paper, we have examined heritage Spanish speakers' crosslinguistic production patterns of /ɲ/ in monolingual Spanish mode and /n+j/ in monolingual English mode to determine whether their phonological systems interact in this scenario. At the group level, we did not find any evidence of interaction and concluded that this group of early Spanish/English bilinguals maintain separate representations for /ɲ/ from /n+j/ based on measures of duration and formant trajectories of the following vocalic portion taken from /ɲ/ data in Spanish mode and /n+j/ in English mode. Comparison with the L2 data suggests that age of acquisition is a likely predictor of interacting systems in this case, at least at the group level. We addressed individual variation in the sample via the relationships between duration and formant contours and dominance and proficiency, and did not find any clear explanatory trends. The next step in this line of investigation will thus be to determine why some bilinguals evidence formant trajectory patterns at the individual level that diverge from the group-level pattern. To that end, we highlighted the need to (a) replicate the study with a larger sample that spans a wider range of proficiency and dominance and (b) test perception to isolate the acoustic cues that are used to distinguish /ɲ/ and /n+j/ in the input. It will also be valuable to directly compare the heritage data with non-heritage native speaker data in the HL to determine how the representations of these populations, who typically differ in dominance and input quantity/quality (among other factors), overlap. Finally, it will be of interest to determine whether language mode plays a role in the interaction of bilinguals' systems when it comes to these sounds. In the current study, we tested in monolingual modes in order to give participants the best chance possible of producing distinct segments. However, research on language mode in HS has shown that a bilingual versus monolingual mode in testing plays a role in both production and perception (e.g., Amengual 2018; Antoniou et al. 2012; Simonet and Amengual 2020). What would happen if were to test these HS in a bilingual mode (which is common for this community, in which participants are able to—and often do—codeswitch between the languages)? Would we see evidence of interaction? If so, what does that tell us about the nature of these representations? Ultimately, the triangulation of data from various bilingual profiles in monolingual and bilingual testing modes will lead us further towards the goal of a holistic understanding of the nature of interacting systems in the bilingual brain.

**Author Contributions:** Conceptualization, S.S. and J.C.; methodology, S.S. and J.C.; software, S.S. (for data), J.C. (for statistical analysis); validation, S.S. and J.C.; formal analysis, S.S. and J.C.; investigation, S.S. and J.C.; resources, J.C.; data curation, S.S.; writing—original draft preparation, S.S. and J.C.; writing—review and editing, S.S. and J.C.; visualization, S.S. and J.C.; supervision, S.S. and J.C.; project administration, S.S. and J.C. All authors have read and agreed to the published version of the manuscript.

**Funding:** This research received no external funding.

**Acknowledgments:** We would like to thank David Abugaber for his assistance with the SSANOVA analyses as well as Leire Echevarria and Brian Rocca for their help with data collection. We also wish to thank Mark Amengual as the Guest Editor of this issue and the two anonymous reviewers for their valuable feedback.

**Conflicts of Interest:** The authors declare no conflict of interest.

# Appendix A

**Table A1.** Stimuli.

|  | Spanish Mode | | English Mode | |
| --- | --- | --- | --- | --- |
| Critical<br>(C)CV.ɲa (Spanish)<br>(C)CVn.ja (English) | reña | [reɲa] | renya | [ɹɛnjə] |
|  | boña | [boɲa] | bonya | [bɑnjə] |
|  | broña | [bɾoɲa] | bronya | [bɹɑnjə] |
|  | droña | [dɾoɲa] | dronya | [dɹɑnjə] |
|  | feña | [feɲa] | fenya | [fɛnjə] |
|  | poña | [poɲa] | ponya | [pʰɑnjə] |
|  | foña | [foɲa] | fonya | [fɑnjə] |
|  | loña | [loɲa] | lonya | [lɑnjə] |
|  | deña | [deɲa] | denya | [dɛnjə] |
|  | beña | [beɲa] | benya | [bɛnjə] |
| Control<br>(C)CV.na | bena | [bena] | benna | [bɛnə] |
|  | dena | [dena] | denna | [dɛnə] |
|  | lona | [lona] | lonna | [lɑnə] |
|  | fona | [fona] | fonna | [fɑnə] |
|  | pona | [pona] | ponna | [pʰɑnə] |
|  | fena | [fena] | fenna | [fɛnə] |
|  | drona | [dɾona] | dronna | [dɹɑnə] |
|  | brona | [bɾona] | bronna | [bɹɑnə] |
|  | quena | [kena] | renna | [ɹɛnə] |
|  | jona | [xona] | bonna | [bɑnə] |
| Distractor | nela | [nela] | talla | [tʰælə] |
|  | neda | [neða̝] | tamma | [tʰæmə] |
|  | dera | [deɾa] | tulla | [tʰʌlə] |
|  | gada | [gaða̝] | bura | [bɚə] |
|  | meba | [meβa̝] | lekka | [lɛkə] |
|  | bera | [beɾa] | meppa | [mɛpə] |
|  | doda | [doða̝] | maffa | [mæfə] |
|  | bora | [boɾa] | ponka | [pʰɑnkə] |
|  | doba | [doβa̝] | cromma | [kʰɹɑmə] |
|  | gora | [goɾa] | neppa | [nɛpə] |
|  | gera | [geɾa] | zappa | [zæpə] |
|  | pada | [paða̝] | ficka | [fɪkə] |
|  | fala | [fala] | vatta | [værə] |
|  | deda | [deða̝] | virta | [vɚˑɾə] |
|  | seba | [seβa̝] | zanta | [zæntə] |
|  | poba | [poβa̝] | thappa | [θæpə] |
|  | dola | [dola] | thurpa | [θɚpə] |
|  | teba | [teβa̝] | drotta | [dɹɑɾə] |
|  | dela | [dela] | vecka | [vɛkə] |
|  | bada | [baða̝] | stucka | [stʌkə] |

## Appendix B

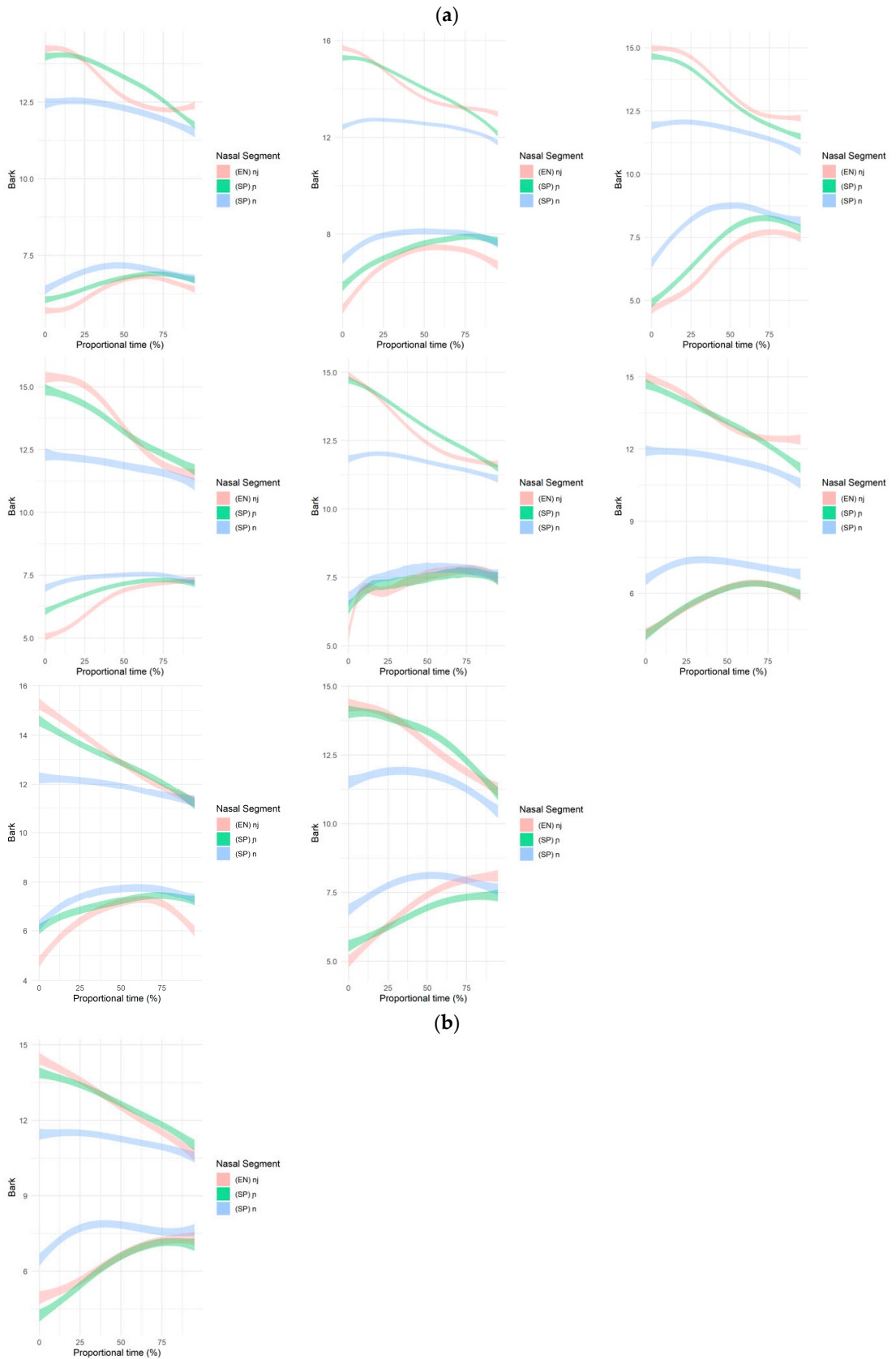

**Figure A1.** *Cont.*

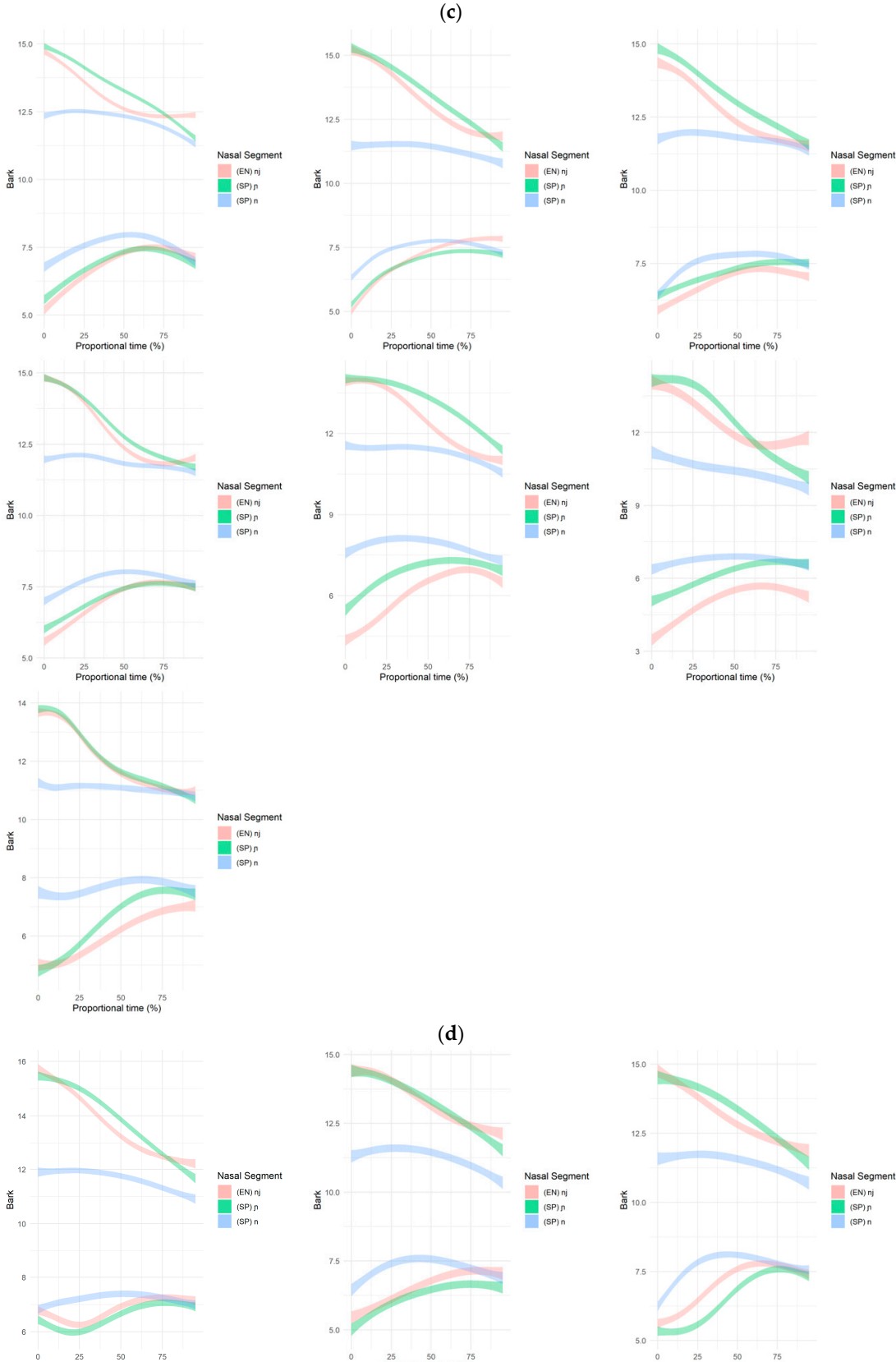

**Figure A1.** (**a**) Pattern 1: individual splines that follow the group pattern. (**b**) Pattern 2: individual splines that differ in F1 from the group pattern. (**c**) Pattern 3: individual splines that differ in F2 from the group pattern. (**d**) Pattern 4: individual splines that differ in F1 and F2 from the group pattern.

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
