# Peer review of "Shared or Separate Representations? The Spanish Palatal Nasal in Early Spanish/English Bilinguals"

_languages, doi:10.3390/languages5040050_

Round 1

Reviewer 1 Report

This is a fascinating study that will serve to further our understanding of bilingual phonology as well as heritage language acquisition. Additionally, the palatal nasal in Spanish has not received as much scholarly attention is it probably should have, so this is an exciting avenue as well.

The manuscript is extremely well written, and I only have a very few small comments. 
p. 1, line 24: the "different flavors" sentence seems a bit out of place, I would find a different metaphor or ad different way to explain that heritage bilingualism falls within the subfield of bilingualism.

p. 3, line 12: I would add a "but" rather than the parentheses, so "a similar but heverosyllabic /n+j/ sequence"

p. 3, footnote 2: typo, should be "in this second sequence in Spanish is not phonemic" (it currently says "is Spanish is")

p. 7, line 259: could you explain, even in just a phrase, why the 5.2% of the data had to be manually corrected (for what, and how)?

p. 11., line 362: this is minor but the paragraph that starts on this line is in a larger font and is centered, so just to keep in mind before final version/typesetting

Author Response

Response to reviewers

We thank both reviewers for their insightful comments, which have no doubt improved the quality of the manuscript, and address each in turn below.

Reviewer 1 Comments:

This is a fascinating study that will serve to further our understanding of bilingual phonology as well as heritage language acquisition. Additionally, the palatal nasal in Spanish has not received as much scholarly attention is it probably should have, so this is an exciting avenue as well.

The manuscript is extremely well written, and I only have a very few small comments.

  1. 1, line 24: the "different flavors" sentence seems a bit out of place, I would find a different metaphor or ad different way to explain that heritage bilingualism falls within the subfield of bilingualism. Thank you; we have changed “flavors” to “forms” and it now reads “bilingualism comes in many different forms.”
  2. 3, line 12: I would add a "but" rather than the parentheses, so "a similar but heterosyllabic /n+j/ sequence" Change made
  3. 3, footnote 2: typo, should be "in this second sequence in Spanish is not phonemic" (it currently says "is Spanish is") Fixed, thank you
  4. 7, line 259: could you explain, even in just a phrase, why the 5.2% of the data had to be manually corrected (for what, and how)? Added “where it was evident that there were formant tracking errors with the Praat script” to the end of this sentence.
  5. 11., line 362: this is minor but the paragraph that starts on this line is in a larger font and is centered, so just to keep in mind before final version/typesetting Fixed, thank you

Reviewer 2 Report

Thank you for the opportunity to read this manuscript. This is research worthy publication. I have one main concern, and some minor details. See my comments in attached document.

Author Response

Response to reviewers

We thank both reviewers for their insightful comments, which have no doubt improved the quality of the manuscript, and address each in turn below.

Reviewer 2 comments:

Major issues

  • This article compares Spanish /ɲ/ and English /n+j/. One big difference between languages is that Spanish does include both representations (though arguably absent in some dialects, Bongiovanni 2019, Kochetov & Colantoni 2011, Malmberg 1950, Moya 1993, Peña 2016a, 2016b) in its inventory: /ɲ/ and /n+j/. Why wasn’t Spanish /n+j/ included in the comparison? This is not a robust contrast even in Spanish, and, there are very few minimal pairs that typically have very low frequency of occurrence. In other words, the /ɲ/-/n+j/ contrast has a low functional low in Spanish. Considering that heritage speakers are usually (very) literate in one language (English, in this case) but not the other, how robust is this contrast for them in Spanish if there may not be any incentive to keep it? Do the heritage speakers surveyed implement a distinction between them in Spanish? Also, is there a cross-linguistic difference (especially considering the difference in their syllabic interpretations) for /n+j/ for this speaker group? While these may not be the main point of the article, it seems to me that the comparison is more complex than as currently presented.

We thank the reviewer for this comment. We address this in Footnote 7, which discusses the potential role of Spanish /nj/. We absolutely agree that Spanish /nj/ data would give a more complete picture of the crosslinguistic relationship (and state this in the fn), and, in light of an absence of these data, have been cautious in the strength of our conclusions. It no doubt would have been ideal to include Spanish /nj/ as a condition in the Spanish task and it will be included in the next phase of this study, but we unfortunately do not have these data. The goal of the paper was to determine whether the speakers maintain a distinction between /ɲ/ in Spanish and /nj/ in English, and now the next steps are to incorporate Spanish /nj/ to identify the crosslinguistic /nj/ representation and with special attention to the syllabic structure of the sequence in each language.

Minor issues

  • If possible, change figures 1-3 so that duration is congruent across spectrograms. It would make visualizations more comparable. While we agree with the reviewer, the spectrograms intentionally each contain the full token to illustrate the transitions, etc.; to our knowledge, the only way to make the duration congruent would be to only present the nasal and following vowel and remove the preceding material. With this reasoning in mind, we hope it is satisfactory to leave them as is and that the durational difference will be sufficiently visibly evident to the reader.
  • Legend in figure 4 says “nasal segment”, but caption and prose make reference to the duration of the vocalic portion. Should the y-axis in figure 4 it read “vocalic portion”? The y axis is categorical and refers to the nasal segment, i.e., whether the boxplot represents /ɲ/ data or /nj/ data. We have added a note to the figure caption to make sure this is clear to the reader.
  • Regarding the talker, were both of them of the same gender? Added in lines 228-229 that both speakers were female.
  • I would suggest changing colors in figures to make them more accessible to color blind readers. Thank you; we agree that making the article as accessible as possible should be a priority wherever possible. To our understanding, the color scheme used is visible to those with protanopia and deuteranopia. With that said, our knowledge in this case is limited and we therefore leave this change up to the editors. If a change needs to be made, our research assistant will need to re-generate all the plots, which will require an additional week’s turnaround time.
  • More than pointing out the adjacent/overlapping contours between 40% and 100% of F1, the key difference between languages in figure 5 is the non-overlapping contours between 0% and 40%. I would suggest highlighting this fact. Thank you. We have now changed the sentence in line 296 so that now it reads “For F1, the confidence intervals of the /ɲ/ and /n+j/ curves do not overlap between the 0% and 40% points, after which they run adjacent to one another between the 40% and 100% points, with a slight overlap between 50% and 75%.”
  • Are the Bark difference (where available) within JND values? We have added the following into the text.

Lines 310-316: “  This difference falls below the assumed just-noticeable difference threshold of 1 Bark unit, which we address in the discussion in terms of whether this difference is perceivable. In contrast, there is zero overlap in the confidence intervals for Spanish /n/ versus /n+j/ and /ɲ/ with differences that exceed the JND threshold. For Spanish /n/ versus English /n+j/ differences in F1 ranges from 1.15 to 3.17 Bark and in F2 from 1.61 to 3.68 Bark at their most different. For Spanish /n/ versus /ɲ/ differences in F1 range from 1.00 to 3.01 Bark and in F2 from 1.48 to 3.16 Bark at their most different.”  

Lines 458-459 “ […] even in the cases in which the contour differs beyond the JND threshold.”

  • Bongiovanni (2015) found that listeners could discriminate /n/ from both /ɲ/ and /n+j/, but when presented with /ɲ/ and /n+j/ d’ values indicated discrimination at random. That is, listeners –who were native speakers of a dialect in which /ɲ/ and /n+j/ have arguably merged, at least among younger speakers– could not tell the difference between accurately discriminate between /ɲ/ and /n+j/. In referencing this study, footnote 7 reads “this distinction is perceivable to L1 Spanish listeners”.

            Thank you very much for bringing this oversight to our attention. We have edited the text from the footnote to note that, even in a situation in which there is a near-merger such as BAs Spanish, that speakers produce acoustically distinct /ɲ/ and /nj/.

•          For the individual data analysis, the authors note that the rise in F2 indicate a more fronted realization. The higher F2 values may also indicate of a more constricted dorsopalatal realization given that dorsopalatal constriction narrowing and F2 are positively correlated, so there is another interpretation for the acoustic facts. Thank you for this alternate possibility- we have added a footnote to line 403 that says “As pointed out by an anonymous reviewer, another interpretation of the higher F2 values is that these speakers could be producing a more constricted dorsopalatal realization given that dorsopalatal constriction narrowing and F2 are positively correlated.